# Decaying quantum turbulence in a two-dimensional Bose-Einstein condensate at finite temperature

Andrew J. Groszek[1, 2★], Matthew J. Davis[3] and Tapio P. Simula[2, 4]

**1** Joint Quantum Centre (JQC) Durham–Newcastle, School of Mathematics, Statistics and Physics, Newcastle University, Newcastle upon Tyne NE1 7RU, United Kingdom
**2** School of Physics and Astronomy, Monash University, Victoria 3800, Australia
**3** ARC Centre of Excellence in Future Low-Energy Electronics Technologies, School of Mathematics and Physics, University of Queensland, Brisbane, Queensland 4072, Australia
**4** Optical Sciences Centre, Swinburne University of Technology, Melbourne 3122, Australia

★ andrew.groszek@newcastle.ac.uk

## Abstract

We numerically model decaying quantum turbulence in two-dimensional disk-shaped Bose–Einstein condensates, and investigate the effects of finite temperature on the turbulent dynamics. We prepare initial states with a range of *condensate temperatures*, and imprint equal numbers of vortices and antivortices at randomly chosen positions throughout the fluid. The initial states are then subjected to unitary time-evolution within the c-field methodology. For the lowest condensate temperatures, the results of the zero temperature Gross–Pitaevskii theory are reproduced, whereby vortex evaporative heating leads to the formation of Onsager vortex clusters characterised by a negative absolute *vortex temperature*. At higher condensate temperatures the dissipative effects due to vortex–phonon interactions tend to drive the vortex gas towards positive vortex temperatures dominated by the presence of vortex dipoles. We associate these two behaviours with the system evolving toward an anomalous non-thermal fixed point, or a Gaussian thermal fixed point, respectively.

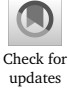
# 1 Introduction

Developing a complete understanding of turbulent dynamics in fluids remains a significant challenge in contemporary physics. Over many decades of research, a wide range of emergent features have been identified in turbulent systems, such as cascades of energy and enstrophy through wavenumber space [1–3]. However, in general such features have proven highly nontrivial to describe from first principles. Recently, quantum turbulence (QT) in superfluids has emerged as a mature research field [4], and is promising insights into many long-standing problems of hydrodynamics [5]. On a microscopic level, the structure of QT is fundamentally different from its classical counterpart, taking the form of a tangled network of quantised, topologically protected vortex filaments. Nonetheless, many results from classical hydrodynamics have already been reproduced in superfluid systems [6–8], including the Kolmogorov $k^{-5/3}$ energy scaling law [9, 10].

In the case of two-dimensional (2D) turbulence, this classical–quantum connection has motivated several studies aimed at realising the inverse energy cascade in 2D QT [11–16]—a well known phenomenon in driven classical 2D turbulence [2, 17]. In 2D superfluid turbulence this phenomenon, as predicted by Onsager's thermodynamic model of point-like vortices [18], should be associated with the clustering of same-sign vortices at negative absolute vortex temperatures [13,15,19–24], as these two phenomena are both characterised by the emergence of system-scale eddies. Indeed, the applicability of Onsager's model to 2D QT is striking—in two recent experiments [25,26], large numbers of vortices were injected into planar Bose–Einstein condensates (BECs) and evidence was obtained for the formation of high-energy Onsager vortex clusters. These experiments have only become possible recently due to advances in the imaging and control of quasi-2D BECs [27,28], as well as the possibility of detecting vortex signs in a turbulent state [29,30].

Previous numerical work on the dynamics of randomly imprinted vortices in planar BECs identified that Onsager vortices could emerge in the ensuing dynamics, using both a Gross–Pitaevskii model and a 2D point vortex model with phenomenological pair annihilation [20]. The mechanism for the Onsager vortex formation was identified as being evaporative heating, where vortex pair annihilation led to an increased incompressible kinetic energy per vortex, and forced the system into the negative absolute temperature region of the vortex phase space. Subsequent analysis showed that Onsager vortex formation was inhibited in harmonically trapped BECs due to the inhomogeneous condensate density [22]. The same authors also found that the inclusion of phenomenological dissipation representing the effects of damping due to finite condensate temperature also had a deleterious effect on the formation of Onsager vortices [22]. These findings raised important questions regarding the possibility of experimentally observing Onsager vortices in decaying two-dimensional quantum turbulence, and motivate

more quantitative studies of the effect of the temperature of the atoms in these systems.

Here we revisit the question of how non-zero condensate temperature affects Onsager vortex formation in decaying two-dimensional quantum turbulence. Rather than incorporating thermal atom effects in the Gross–Pitaevskii equation (GPE) using a phenomenological damping term, we instead perform dynamical simulations using the classical field methodology [31–33]. Briefly, this uses the Gross-Pitaevskii equation to simulate the dynamics of not only the condensate, but also the low-energy thermal fluctuations of the field. We simulate the grand canonical stochastic projected Gross–Pitaevskii equation (SPGPE), describing the classical field coupled to a bath, to generate initial thermal ensembles [33,34]. We imprint vortices on these microstates to form an ensemble of vortex distributions, and determine the resulting effect of the finite temperature on the turbulent vortex dynamics by integrating the microcanoncial projected Gross–Pitaevskii equation (PGPE) that conserves both energy and normalisation of the classical field.

In our simulations we focus on low temperature finite-size systems in which thermal activation of vortex-antivortex pairs is suppressed [35]. We quantify the dynamics using the vortex thermometry methodology [24] facilitated by vortex classification techniques [13,36]. Our results show that for sufficiently cold condensates, there is little difference from the predictions of the zero temperature GPE. Furthermore, our results suggest that the vortex evaporative heating mechanism overwhelms the dissipative effects due to thermal atoms for condensate fractions above approximately 80% for our model, which is an experimentally attainable regime.

This paper is organised as follows: in Section 2 we give a brief overview of the c-field methodology, including the initial state preparation at finite temperature and the numerical techniques employed. Section 3 presents the numerical results for the decaying quantum turbulence at finite temperature, and uses vortex thermometry to analyse the dynamics of the vortex subsystem. Evidence is also provided for universal scaling in our simulations, and based on this we are able to interpret the dynamics as evolving towards either a thermal or non-thermal fixed point, depending on the temperature of the system. Finally, we discuss the results and conclude in Section 4.

## 2  c-field modelling of finite temperature Bose-Einstein condensates

In this work we model the dynamics of a partially-condensed Bose gas at finite temperature using a projected [32,33,37] and stochastic projected [33,34,38] Gross–Pitaevskii equation. Our numerical modelling consists of two distinct stages. We first prepare a number of statistically equivalent initial conditions for the Bose field confined by a two-dimensional disk trap at a given condensate temperature, varying both the spatial noise distribution (generated by evolving the SPGPE) and the initial vortex configuration (randomly imprinted into the field) for each state. We then perform microcanonical evolution for each of the initial states using the energy conserving PGPE. We outline these steps in detail below.

### 2.1  Initial state preparation

We begin by finding the ground state of the trapping potential, $V_{\mathrm{tr}}(\mathbf{r})$, using imaginary time propagation of the zero-temperature projected Gross–Pitaevskii equation,

$$i\hbar \frac{\partial \psi(\mathbf{r}, t)}{\partial t} = \mathcal{P}\{L_{\mathrm{GP}}\psi(\mathbf{r}, t)\}. \tag{1}$$

Here $\psi(\mathbf{r}, t)$ is the classical Bose field that includes only the highly-occupied, low momentum modes of the gas below a chosen cutoff. The Gross-Pitaevskii operator is

$$L_{\text{GP}} = -\frac{\hbar^2}{2m}\nabla^2_{\text{2D}} + V_{\text{tr}}(\mathbf{r}) + g|\psi(\mathbf{r}, t)|^2, \tag{2}$$

where $m$ is the atomic mass, and $g = 4\pi\hbar^2 a_s N/ml_z$ is the two-dimensional interaction constant describing the strength of the $s$-wave interatomic collisions (with scattering length $a_s$) for system with (uniform) axial extent $l_z$ and total atom number $N$. The projection operator $\mathcal{P}$ ensures that no population is transferred to the higher momentum modes in the numerics. From here on, the time dependence of the classical field is implied, $\psi(\mathbf{r}) \equiv \psi(\mathbf{r}, t)$.

To numerically represent a uniform disk geometry, we use a potential $V_{\text{tr}}(r) = \mu_\circ(r/R)^{30}$ with trap radius $R \approx 60\,\xi_\circ$, where $\mu_\circ$ is the chemical potential of the ground state, and $\xi_\circ = \hbar/(2m\mu_\circ)^{1/2}$ is the corresponding healing length (which is on the order of the vortex core size). We set $g = 5250\,\hbar^2/m$, which for a gas of $10^5$ $^{87}$Rb atoms would correspond to a cloud with $l_z \approx 1.3\,\mu\text{m}$. Throughout this work, we adopt units of $\mu_\circ$ (energy), $\xi_\circ$ (length), $t_\circ \equiv \hbar/\mu_\circ$ (time) and $T_\circ \equiv \mu_\circ/k_B$ (fluid temperature). If the physical radius of the disk trap is fixed at $R = 30\,\mu\text{m}$, then these units would correspond to $\mu_\circ \approx k_B \times 11\,\text{nK}$, $\xi_\circ \approx 0.5\,\mu\text{m}$ and $t_\circ \approx 0.68\,\text{ms}$.

The effects of finite condensate temperature are incorporated by subsequently evolving the classical Bose field using the stochastic projected Gross–Pitaevskii equation. This is a grand canonical theory, where the high energy single-particle modes above the cutoff are treated as a thermalised reservoir, with well-defined temperature $T$ and chemical potential $\mu$, that exchanges particles and energy with the classical field [33]. In Stratonovich form, the SPGPE is expressed as:

$$d\psi(\mathbf{r}) = \mathcal{P}\left\{-\frac{i}{\hbar}L_{\text{GP}}\psi(\mathbf{r})dt + \frac{\gamma}{\hbar}(\mu - L_{\text{GP}})\psi(\mathbf{r})dt + dW(\mathbf{r}, t)\right\}, \tag{3}$$

where $\gamma$ is a dimensionless growth coefficient, and the complex noise term $dW(\mathbf{r}, t)$ is spatially and temporally uncorrelated, with its only non-zero moment given by $\langle dW^*(\mathbf{r}, t)dW(\mathbf{r}', t)\rangle = (2\gamma k_B T/\hbar)\delta(\mathbf{r} - \mathbf{r}')dt$. The first term on the right hand side of Eq. (3) corresponds to unitary evolution of the field, while the second and third terms model condensate growth processes resulting from collisions of atoms above the momentum cutoff enforced by $\mathcal{P}$ [33, 34, 38].

To generate a thermalised classical field at a chosen non-zero temperature $T$, we evolve the SPGPE using a growth coefficient of $\gamma = 10^{-2}$ (this choice is arbitrary and does not affect the final equilibrium state). The reservoir chemical potential is chosen to be $\mu = \mu_\circ$, and the temperature is set to one of three values, $T/T_0 \approx \{0.9, 1.8, 2.7\}$, for each simulation ensemble. Note that we also perform a $T = 0$ simulation, but we do not need to evolve the SPGPE (3) to find the initial state. For comparison, the critical temperature for condensation in this system is $T_c \approx 10\,T_\circ$ which we find by estimating the temperature at which the condensate fraction of the classical field vanishes in equilibrium while holding the chemical potential constant (see below for details regarding the condensate fraction measurement). Note that this is different to the typical scenario in which the total number of atoms is kept constant as the temperature is varied.

After an initial burn-in time of $\sim 200\,t_\circ$, the norm $\int|\psi(\mathbf{r})|^2 d\mathbf{r} \approx 1$ (which is weakly temperature-dependent) and the total energy attain approximately stable values (with fluctuations $\lesssim 1\%$), indicating that equilibrium has been reached. For each chosen temperature, 50 uncorrelated samples of the stochastically evolving field are used as thermalised initial conditions.

At a given temperature, the condensate fraction $n_0$ can be determined from a large number of uncorrelated states by applying the Penrose–Onsager criterion [39], whereby $n_0$ is identified as the largest eigenvalue of the one-body density matrix, $\rho(\mathbf{r}, \mathbf{r}') = \langle \psi^*(\mathbf{r}) \psi(\mathbf{r}') \rangle$, with the average taken over different stochastic realisations. The condensate mode $\psi_0(\mathbf{r})$ is then given by the corresponding eigenvector. For our chosen temperatures, the measured condensate fraction ranges between $0.75 \lesssim n_0 \lesssim 1$. When making comparisons with experiments it is important to keep in mind that these values are only an approximation to the true condensate fraction, as thermal atoms with momenta above the cutoff are not included in the classical field.

For each of the sampled initial conditions, vortices are then imprinted by multiplying the field $\psi(\mathbf{r})$ by an ansatz function $\eta(\mathbf{r})$, which establishes both a density dip and a $2\pi$ phase-winding around each chosen vortex core location. The ansatz is defined

$$\eta(\mathbf{r}) = \prod_{k=1}^{N_v} \chi(\mathbf{r} - \mathbf{r}_k) \exp\left[ i s_k \arctan\left( \frac{y - y_k}{x - x_k} \right) \right], \tag{4}$$

where $N_v$ is the number of vortices being imprinted, and $\mathbf{r}_k = (x_k, y_k)$ and $s_k \in \pm 1$ are the position and sign, respectively, of the $k$th vortex. The real function $\chi(\mathbf{r}) = r/(r^2 + 2\xi_\circ^2)^{1/2}$ approximates the density profile of each vortex core [40]. We imprint $N_v = 100$ randomly distributed vortices, with equal numbers of each sign to ensure that the angular momentum of the condensate remains close to zero. After the vortices have been imprinted, the field is normalised to its initial value to avoid a net loss of probability density. The vortex imprinting adds a small amount of kinetic energy to the system and therefore slightly increases the temperature of the field.

## 2.2 Microcanonical evolution

After vortex imprinting, the turbulent dynamics are simulated by evolving each state using the PGPE for $t \approx 5500\,t_\circ$ (corresponding to $t \approx 3.8\,\text{s}$ for the physical parameters chosen in Sec. 2.1). The phonons of the field interact with the vortices, causing additional damping and changing the nature of the decaying turbulence. During the evolution we track the positions of the vortices over time by locating phase singularities in the classical field. To avoid spurious vortex detections due to density fluctuations at high temperatures, we first coarse-grain the field by removing spatial frequencies beyond $\pi/\xi_\circ$ before performing the vortex detection step. Additionally, we only count vortices within the region $r < 0.95\,R$, in order to avoid the detection of numerical 'ghost' vortices in regions of low classical field density [41].

## 2.3 Numerical details

We represent the field on a numerical grid of size $(512)^2$, and perform temporal evolution using a fourth-order adaptive Runge–Kutta technique in the software package XMDS2 [42]. The spatial resolution is set such that the grid spacing $\Delta x \approx \xi_\circ/3.5$, which ensures that the vortex cores are accurately represented by the numerics. This choice leads to a value for the wavevector $k_{\text{cut}}$ of the projector. Numerically, we implement the projector $\mathcal{P}$ in Fourier space, where it takes the form of a binary mask which has a value of unity for $|\mathbf{k}| < k_{\text{cut}}$, and is zero outside this region. This prevents occupation of any modes of $\psi(\mathbf{r})$ beyond a wavenumber of $k_{\text{cut}}$. To ensure that the field is de-aliased, we set the cutoff for the projector to $k_{\text{cut}} = k_{\text{max}}/2$, where $k_{\text{max}} = \pi/\Delta x$ is the largest wavenumber that can be represented on our numerical grid.

The c-field method is often considered to be valid when the mode occupations are significantly larger than one $n_k \gg 1$, as this is when these modes can be expected to behave classically [33]. We note, however, that several authors have weakened this condition to $n_k \gtrsim 1$

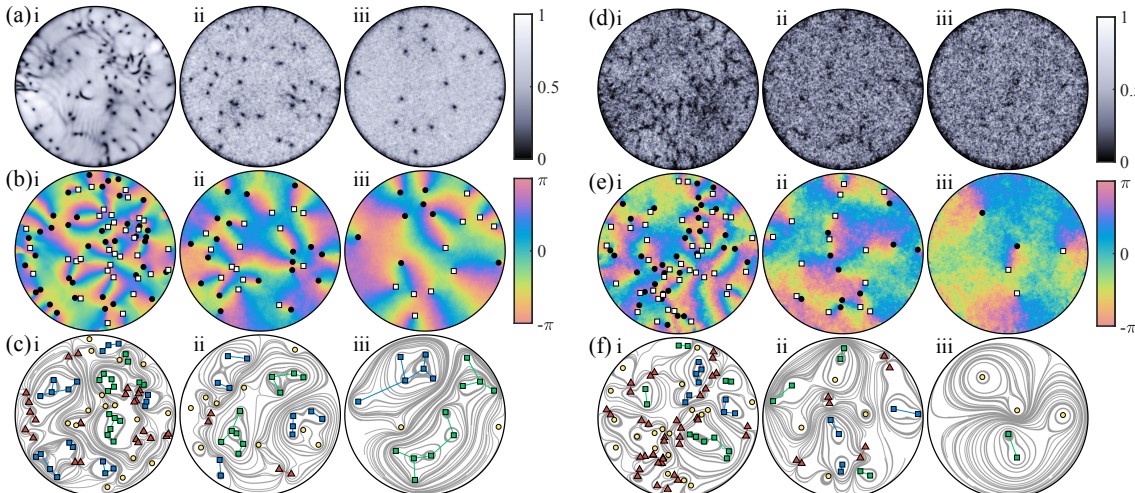

Figure 1: Temporal evolution of the classical field $\psi(\mathbf{r})$ for two condensate fractions, $n_0 = 1$ [(a)–(c)] and $n_0 \approx 0.75$ [(d)–(f)], with frames i, ii and iii in each row corresponding to times $t/t_o \approx 10$, $t/t_o \approx 900$ and $t/t_o \approx 4500$, respectively. For each case the classical field density $|\psi(\mathbf{r})|^2$ is shown in rows (a) and (d), the phase in (b) and (e), and the classified vortices and incompressible velocity field streamlines in (c) and (f). In (b) and (e), vortices (antivortices) are indicated with black dots (white squares), while in (c) and (f), clusters of vortices (antivortices) are identified as blue (green) squares, dipoles as red triangles, and free vortices as yellow circles. In rows (a) and (d), the density is normalised to its maximum value, $1.7 \times 10^{-4}\, \xi_o^{-2}$ and $3.0 \times 10^{-4}\, \xi_o^{-2}$, respectively.

with no apparent ill effects [33]. Here, we set the cutoff based on the numerical grid size, as described above. We then confirm *a posteriori* that this choice is reasonable by calculating the occupation at the cutoff throughout our simulations, and ensuring that it satisfies $n_k(k_{\text{cut}}) \gtrsim 1$ for an experimentally realistic system. For reference, the bare momentum mode occupations for our lowest and highest temperature simulations are shown in the insets of Fig. 4(c,d). One might argue that it would be physically appropriate to increase $k_{\text{cut}}$ with increasing temperature to keep the same occupation number at the cutoff. However, as our main goal is to understand the effect of temperature on the formation of Onsager vortices, we choose to keep all other numerical parameters the same between the different sets of simulations.

## 3 Results

### 3.1 Evaporative heating of vortices

Figure 1 shows exemplary simulation results of decaying two-dimensional quantum turbulence. Figure 1(a)–(c) are for a system at zero condensate temperature and condensate fraction $n_0 = 1$, and Fig. 1(d)–(f) are at finite temperature with a condensate fraction of $n_0 \approx 0.75$. Each frame (a)–(f) shows three snapshots (i–iii) from the simulated dynamics with time increasing from left to right. The top rows show the condensate density, with vortices visible as dark spots. The middle rows show the phase of the classical field, with the locations of vortices and antivortices denoted by black circles and white squares, respectively. The bottom rows show the vortices after they have been classified as same-sign clusters (blue/green markers), vortex dipoles (red markers) and free vortices (yellow markers) [13, 24, 36], as well as the streamlines of the incompressible velocity field of the condensate, which have been approximated using a

point-vortex model [20,43].

In Figure 1(a), a high frequency phonon field develops over time due to the vortex–sound interactions, although these density oscillations have low amplitude. By contrast, at higher condensate temperatures [panel (d)] the density fluctuations are much more prominent, and hence the visibility of the vortex cores is reduced significantly. The vortices are also observed to decay much faster at these temperatures due to the dissipative effect associated with the vortex–phonon interactions.

In cold condensates (a)–(c) the vortex evaporative heating mechanism drives the vortex gas towards states with higher incompressible kinetic energy per vortex, resulting in the formation of Onsager vortex clusters [20, 22] [most evident in panel (c)iii]. In warmer condensates (d)–(f) the vortex cooling effect due to the dissipative interaction with the non-condensate atoms overwhelms the vortex evaporative heating mechanism, and hence the formation of Onsager vortex clusters is suppressed.

## 3.2    Vortex thermometry

In statistical equilibrium a 2D vortex gas can be described by a vortex temperature [18,20,24], which determines the incompressible kinetic energy of the flow field. The inverse temperature is defined as $\beta = (1/k_B)\partial S/\partial E$, where $S$ and $E$ are the entropy and energy of the vortex gas, respectively. In the uniform disk system considered here, there are three distinct equilibrium phases of the neutral vortex gas, dependent only on the configuration of the vortices and antivortices in the system:

 (i) At positive temperatures ($\beta \gg 0$), the vortices pair into tightly bound vortex–antivortex dipoles in order to reduce the energy in the flow field they produce.

 (ii) For $\beta \approx 0$, the vortices distribute themselves randomly throughout the system, resulting in a velocity field with an intermediate energy.

 (iii) At negative temperatures ($\beta \ll 0$), the vortices arrange into two large clusters (one of each circulation sign), thereby creating high energy, system-scale rotational flows. These negative absolute temperature vortex states were first predicted by Onsager [18], who realised that the phase space for vortices in a bounded container is restricted, leading to a decrease in the entropy at high energies, and therefore a negative value of $\partial S/\partial E$.

Here, we are able to determine the vortex temperature $\beta$ directly from our simulations by monitoring the fractional populations of classified vortex dipoles and clusters [36]. This is possible because these populations both vary monotonically with temperature in thermodynamic equilibrium, and thus can be used as thermometers as shown previously by Groszek *et al.* [24].

In Fig. 2, the fractional populations of vortices belonging to (a) vortex clusters and (b) vortex dipoles are shown as a function of time, in addition to (c) the inverse vortex temperature $\beta$ determined from the clustered fraction [24]. At zero condensate temperature, the clustered fraction grows fairly monotonically as the evaporative heating of the vortex gas proceeds [panel (a)], while the dipole fraction decays correspondingly [panel (b)]. These trends indicate that the vortex gas is evolving towards states with higher energy per vortex [20]. By contrast, at the highest condensate temperature ($n_0 \approx 0.75$), the clustered (dipole) fraction shows a decreasing (increasing) trend, corresponding to a more rapid loss of incompressible kinetic energy over time.

For all condensate fractions the vortex temperature [Fig. 2(c)] begins near $\beta = 0$, and shows evidence of initial evaporative heating of vortices (towards negative $\beta$). However, in all cases, there is a turning point at which the gradient of $\beta(t)$ changes sign and the vortex system begins to cool. The timescale at which this occurs decreases with increasing condensate temperature,

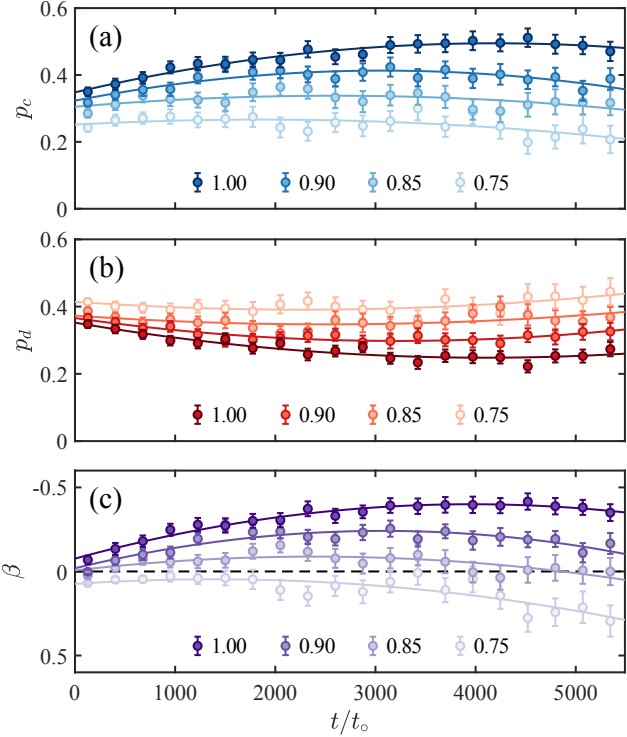

Figure 2: Fractional population curves of (a) vortex clusters and (b) vortex dipoles, as well as (c) the resulting inverse vortex temperature $\beta$, measured using the cluster population [24]. All data points have been ensemble averaged over 50 stochastic realisations, as well as temporally binned, and the error bars correspond to the standard error in the mean of each measurement. The legend in each subfigure indicates the condensate fraction, and the solid lines shown are quadratic fits to the data, serving as a guide to the eye.

indicating that the rate of dissipation of vortex energy into sound increases for higher initial condensate temperatures. Note that, in this figure, positive temperatures are scaled with respect to the Berezinskii–Kosterlitz–Thouless transition temperature [44–47], $\beta_{\mathrm{BKT}} = E_\circ/2$, while the negative temperatures are expressed in terms of the Einstein–Bose vortex condensation temperature [36, 44, 48], $\beta_{\mathrm{EBC}} = N_v E_\circ/4$, where the constant $E_\circ = \rho_s \kappa^2/4\pi$ is defined in terms of the superfluid density $\rho_s$ and the quantum of circulation $\kappa = h/m$.

## 3.3  Vortex number decay

As the fluid (vortex–phonon system) relaxes toward equilibrium, the number of vortices, $N_v(t)$, gradually decays due to vortex–antivortex annihilations—mostly within the bulk of the condensate, but also occasionally at the boundary [22]. This number decay behaviour has been a topic of recent interest [22, 49–55] and several attempts have made to describe the decay process using phenomenological rate equations. A consensus seems to be developing that three-vortex (or even four-vortex) events significantly affect the observed dynamics [22, 53–55]; however, the precise form of the rate equation is still a topic of debate.

We have previously proposed a vortex number decay law of the form [22]

$$\frac{\mathrm{d}N_v}{\mathrm{d}t} = -\Gamma_1 N_v - \Gamma_2 N_v^2 - \Gamma_3 N_v^3 - \Gamma_4 N_v^4, \tag{5}$$

where each $\Gamma_n$ term on the right hand side of the equation is interpreted as an $n$-body decay rate—i.e. the rate at which $n$ vortices will collide and lead to an annihilation event (of at least

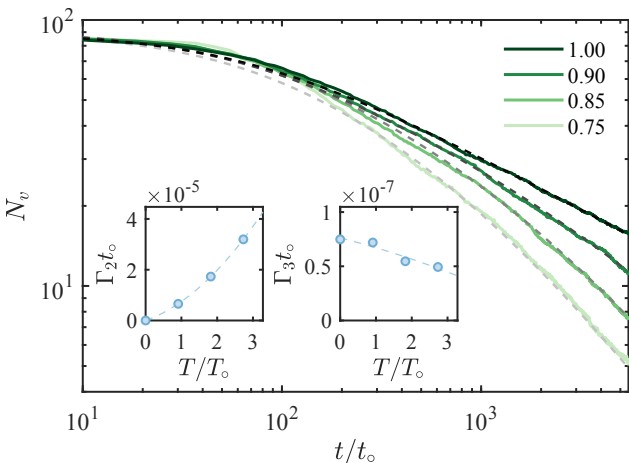

Figure 3: Ensemble averaged vortex number as a function of time for all four condensate temperatures (solid green lines), with fits to Eq. (6) (dashed grey lines). The insets show the measured decay constants as functions of the reduced temperature, with quadratic fits (dashed lines) included as a guide to the data.

two of those vortices). In this interpretation, $\Gamma_1$ is the rate at which vortices leave via the boundary, as single-vortex loss is topologically prohibited in the fluid bulk.

Karl and Gasenzer [54] recently suggested that the decay should instead be modelled using

$$\frac{\mathrm{d}N_v}{\mathrm{d}t} = -\Gamma_2 N_v^2 - \Gamma_3 N_v^{7/2}, \tag{6}$$

where the extra factor of $N_v^{1/2}$ in the three-body term accounts for the vortex-density–dependent velocity of the vortices, which should affect the probability of three-vortex encounters. The one-body term was omitted in their model as their calculations involved an unbounded fluid.

In Fig. 3, we plot the vortex number $N_v$ as a function of reduced time $t/t_\circ$ (solid green lines) for the four different condensate fractions. When fitting the two rate equations (5) and (6), we find that both describe the data equally well. However, since the latter involves fewer free parameters, we opt to use it over our earlier model. The resulting fits are shown as dashed lines in Fig. 3, and the corresponding values of the two fitting parameters are shown as a function of condensate temperature in the insets. Physically, Eq. (6) supports the interpretation that three-body annihilations are significantly more frequent than four-body events, as had been previously argued in Ref. [22]. If the observed $\Gamma_3(T)$ trend were to continue, our data suggest that the three-body term should become negligible at $T \sim 6\,T_\circ$, and lower-order terms would dominate. We note that a one-body term would eventually have to be added to the model at higher condensate temperatures to describe the increasingly steep $N_v(t)$ gradient.

An additional complicating factor in interpreting the physics captured by these phenomenological rate equations arises due to the fact that the decay 'constants' $\Gamma_n(N_v(t))$ are actually time-dependent, as the dominant microscopic decay dynamics are drastically different depending on the density and configuration of the vortex system. It is therefore desirable to find alternative and complementary ways of describing the statistical evolution of the vortices and the system as a whole.

## 3.4 Evidence of universal scaling dynamics

Despite the complexity of 2D QT, it has been demonstrated that the dynamics can in many cases be characterised in terms of statistically steady distributions that are only weakly dependent on the microscopic details of the system. This is a general and powerful approach to understanding

the evolution of a system out of equilibrium, and allows its characterisation in terms of far from equilibrium universality classes [56, 57].

In the following, we demonstrate that in our simulations, both the fluid as a whole, and the vortex distribution embedded therein, display time-invariant statistical behaviour. By studying the growth of correlations as a function of temperature, we are also able to provide some indication as to the eventual fate of the vortex subsystem, interpreted in the context of fixed points as proposed in Ref. [54].

### 3.4.1 Universal dynamics of the field

In order to determine whether the system is undergoing universal dynamics, we look for self-similar evolution in the statistical properties of the field $\psi(\mathbf{r})$, as predicted by the dynamical scaling hypothesis [58]. According to this conjecture, a system undergoing universal evolution displays statistically unchanged behaviour in time following a rescaling that is dependent only on the correlation length $L_c(t)$. Here, $L_c(t)$ characterises the extent over which phase coherence has developed at a given time. We first calculate the two-point correlation function at each time, defined

$$G(\mathbf{r}, t) = \frac{\langle \psi^*(\mathbf{r}+\mathbf{r}', t)\psi(\mathbf{r}', t)\rangle}{\sqrt{\langle |\psi(\mathbf{r}+\mathbf{r}', t)|^2\rangle \langle |\psi(\mathbf{r}', t)|^2\rangle}}, \tag{7}$$

where the angular brackets denote an average taken over both the co-ordinate $\mathbf{r}'$ and all statistical realisations of the field at time $t$. If scaling is occurring, we expect that the correlation

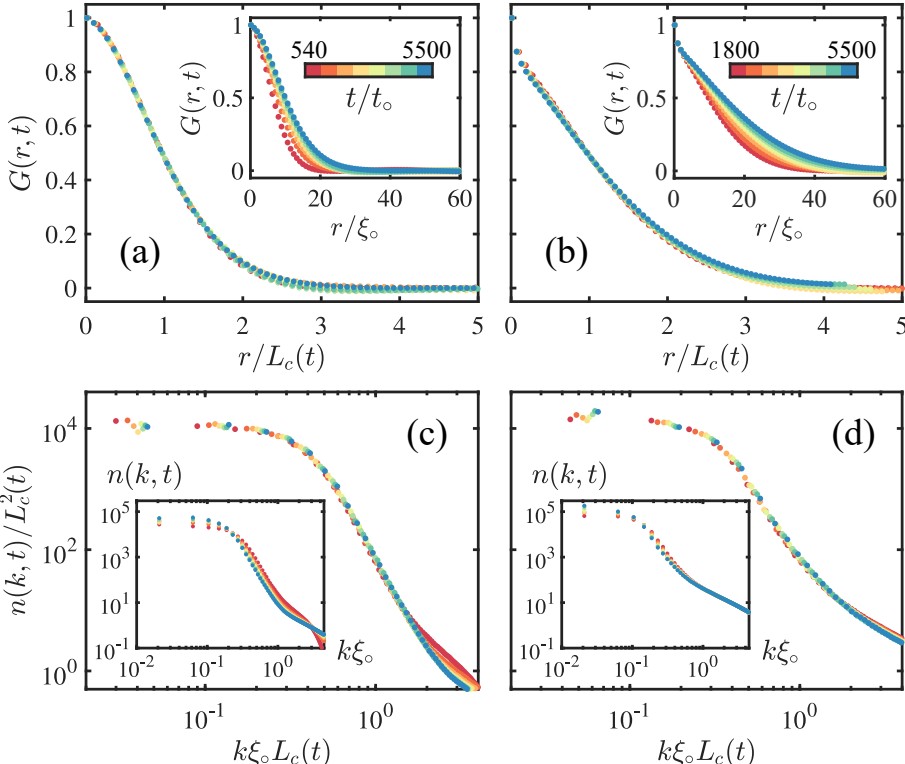

Figure 4: Collapse of correlation functions $G(r, t)$ [(a) and (b)] and occupation number distributions $n(k, t)$ [(c) and (d)] upon rescaling distances with the correlation length $L_c(t)$ (insets show raw data, while main frames show the same curves after rescaling). The left (right) column shows the results for $n_0 = 1$ ($n_0 \approx 0.75$), sampled at eight equally spaced times over the range indicated in the color bar in (a) [(b)]. In (c) and (d), the total number of particles has been normalised to $10^5$.

function will satisfy $G(r,t) = G_{eq}(r)F(r/L_c(t))$, where $G_{eq}(r) = G(r, t \to \infty)$ is the equilibrium correlation function of the system, and $F$ is a universal scaling function [58]. In Fig. 4(a,b), we plot the azimuthally averaged correlation function $G(r,t)$ for a number of sampled times at two of our chosen condensate fractions ($n_0 = 1$ and $n_0 \approx 0.75$, respectively), demonstrating a collapse of the data after rescaling $r \to r/L_c(t)$, and providing evidence that our system is indeed exhibiting universal dynamical scaling. Here, the correlation length has been extracted at each time by determining the radius at which the correlation function falls to a value of 0.5, i.e. $G(L_c(t), t) = 0.5$. The temporal windows shown have been chosen to give the longest range over which a collapse could be obtained. We see similar collapses for the intermediate condensate fractions, $n_0 \approx 0.95$ and $n_0 \approx 0.85$ (data not shown).

We also observe scaling in the azimuthally averaged occupation number spectrum, $n(k,t) = \langle |\hat{\psi}(\mathbf{k},t)|^2 \rangle$ (with $\hat{\psi}$ the spatial Fourier transform of $\psi$), which is predicted to have the form $n(k,t) = L_c^2(t)f(kL_c(t))$, where $f$ is another universal function [58]. The raw and rescaled data are respectively shown in the main frame and inset of Fig. 4(c,d), for the same two temperatures and temporal windows as in Fig. 4(a,b). The rescaled spectra show a reasonable collapse, providing further evidence for universal dynamics of the classical field.

### 3.4.2 Universal dynamics of the vortex configuration

Previously, Groszek *et al.* [24] identified scale invariant behaviour in decaying quantum turbulence by expressing the populations of vortex clusters and dipoles in terms of the total number of vortices remaining in the system, effectively removing the time dependence inherent to the vortex number decay. This analysis revealed that the vortex configuration rapidly approaches a quasiequilibrium steady state characterised by the emergence of power-law distributions. In such a state, it was argued that the vortices must have enough time between each annihilation event to rearrange themselves into the maximum entropy configuration available at their energy. Hence, there are two relevant rates that determine whether the vortex gas has reached quasiequilibrium. The first is the rate at which vortex–antivortex annihilation events are occurring, defined at a given time as

$$\nu_{\mathrm{ann}} \equiv -\frac{1}{2}\frac{dN_v}{dt}, \tag{8}$$

where the factor of one-half accounts for the two vortices lost per annihilation. The second is the rate at which the vortices reconfigure themselves, which we approximate as the inverse of the mean time it takes for a vortex to travel the distance to its nearest neighbour, i.e. $\nu_{ij} \approx \bar{u}/d_{ij}$. Here, $\bar{u} \approx (\hbar/m)(1/d_{ij})$ is the mean velocity of a vortex in a configuration with mean intervortex spacing $d_{ij} \approx R/N_v^{1/2}$. Hence,

$$\nu_{ij} \approx \frac{\hbar N_v}{mR^2}. \tag{9}$$

When $\nu_{\mathrm{ann}} < \nu_{ij}$, the annihilations should be infrequent enough for the system to reach the aforementioned steady state. The two rates can be measured directly from the $N_v(t)$ curves in Fig. 3, and we compare the results in Fig. 5(a) for each initial condensate temperature as a function of vortex number[1]. The point at which quasiequilibrium is reached in each case is indicated with a grey dot.

Figure 5(b) shows the measured cluster $N_c$ and dipole $N_d$ populations as functions of the total vortex number $N_v$ in the system for the four initial condensate temperatures, where the curves have been vertically offset for clarity. Once the state of quasiequilibrium has been reached, the curves are well approximated as power-laws, as seen previously [24]. The exponents at

---

[1]Note that we have calculated $\nu_{\mathrm{ann}}$ using the fits to Eq. (6), rather than the raw $N_v(t)$ data, in order to eliminate noise arising from numerical differentiation.

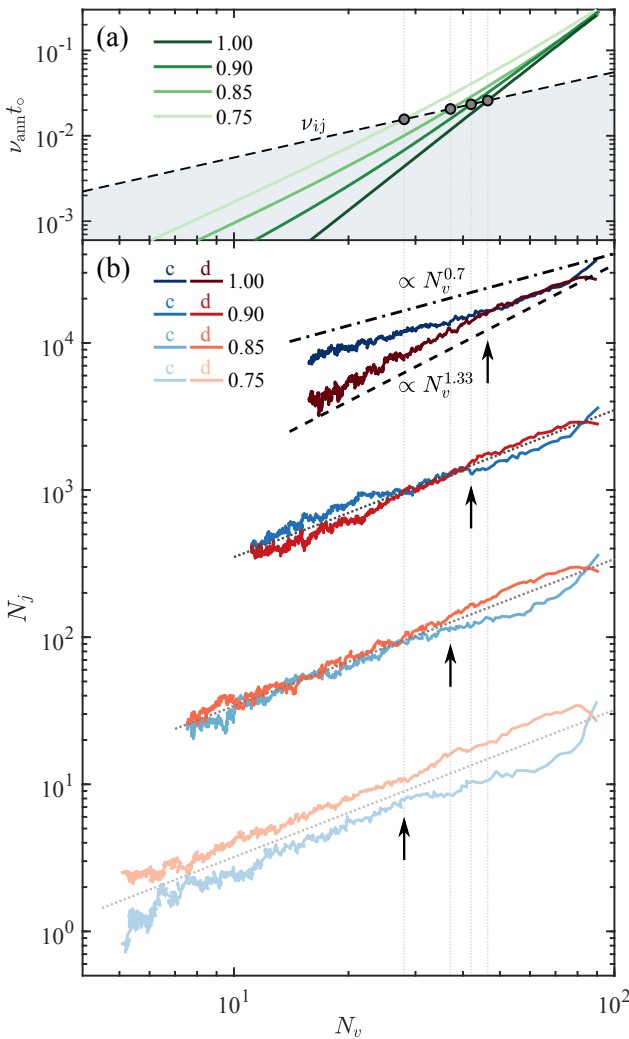

Figure 5: (a) The vortex annihilation rates $v_{\mathrm{ann}}$ for all four initial condensate temperatures (solid green lines) are compared to the rate $v_{ij}$ of vortex reconfiguration (dashed line) as a function of the total vortex number. The shaded region indicates quasiequilibrium, where $v_{\mathrm{ann}} < v_{ij}$, and the black dots denote the points at which this condition is first met for each condensate temperature. (b) The mean number of vortex clusters ($j = c$, blue) and vortex dipoles ($j = d$, red) as functions of total vortex number $N_v$ for each initial condensate temperature. For clarity, the curves are shifted vertically by multiplying $N_j$ by $10^n$, where $n = \{0, 1, 2, 3\}$ for $n_0 = \{0.75, 0.85, 0.90, 1.00\}$, respectively. For $n_0 = 1.00$, the data compare well with $N_c \sim N_v^{0.7}$ (dot–dashed line) and $N_d \sim N_v^{1.33}$ (dashed line). For all other temperatures, $N_j \sim N_v$ (dotted lines) yields a more reasonable comparison. The vertical dotted grey lines are traced from the quasiequilibration points identified in panel (a), and these points are highlighted on each appropriate $N_j$ curve by vertical arrows. Note that time flows right to left in this figure.

zero temperature are consistent with $N_c \sim N_v^{0.7}$ (dot–dashed line) and $N_d \sim N_v^{1.33}$ (dashed line), while at the other condensate temperatures $N_c \sim N_d \sim N_v$ (dotted lines). All four cases show a clear distinction between early- and late-time behaviour, with the cross-over aligning well with the time at which $v_{\mathrm{ann}} = v_{ij}$ (see the arrows in the figure). At early times (high vortex density), the number of dipoles is always greater than or equal to the number of clusters,

before the decay transitions to the late time behaviour, which depends strongly on the fluid temperature. There is also a very early cross-over from cluster-dominated to dipole-dominated behaviour (at $N_v \approx 90$ in each case), which results from the quench-like initial condition. These curves suggest that in the limit of vanishing total vortex number ($N_v \rightarrow 0$), the ultimate fate of the vortex system would be 100% clusters in the two coldest condensate temperature systems and 100% dipoles in the two hottest condensate temperature systems (assuming no further changes to the statistical behaviour would occur).

These two final states act as attractors for the turbulent evolution, and can be interpreted as fixed points at which the dynamics of the system critically slow down, and universal scaling laws emerge [59, 60]. In the case where the system becomes dipole-dominated, the attractor is a Gaussian fixed point, which corresponds to a state where all vortices have annihilated and the fluid thermalises. By contrast, when the vortices form same-sign clusters, annihilation events become infrequent, and the system becomes 'stuck' in a non-equilibrium configuration for an extended period of time. Such behaviour corresponds to an anomalous non-thermal fixed point. It is predicted that the system would eventually return to the Gaussian fixed point, because vortex–phonon interactions give rise to gradual vortex diffusion [61], which serves to break up vortex clusters and encourage vortex–antivortex annihilation. However, it is possible that the time required for the system to cross over to the Gaussian fixed point may approach infinity as $T \rightarrow 0$.

### 3.4.3 Growth of the correlation length

If dynamical scaling is occurring, the correlation length is predicted to grow as a power-law in time, $L_c(t) \sim t^{1/z}$, where $z$ is the dynamical critical exponent. Karl and Gasenzer [54] recently performed simulations of decaying quantum turbulence, and demonstrated that the evolution towards each of the two aforementioned fixed points could be characterised by the observed exponent $z$. They found that, if the system was evolving towards the non-thermal fixed point, then $z \approx 5$; whereas if the system was approaching the Gaussian fixed point, then $z \approx 2$. In that work, the mean nearest-neighbour vortex distance, which we define here as $d_{nn} \equiv \sum_j \min_{k \neq j} |\mathbf{r}_k - \mathbf{r}_j|/N_v$, was used as a measure of the correlation length.

Here, we calculate both $d_{nn}(t)$ and $L_c(t)$ for all temperatures, and plot these two observables as a function of time in Fig. 6 [(a) and (b), respectively]. At late times, we fit a power-law to each curve (shown as dashed grey lines), thereby obtaining two estimates of the exponent $z$ at each temperature, as shown in the inset of (b). The fitting regions are obtained by finding the best region of collapse for $G(r, t)$ (see Sec. 3.4.1). At low temperatures, the two lengths $d_{nn}(t)$ and $L_c(t)$ both show scaling characterised by $z \approx 5$, consistent with evolution towards the non-thermal fixed point. As the temperature is increased, this exponent is found to decrease for both observables, and for the highest temperature, $d_{nn}(t)$ is well described by $z \approx 2$, in agreement with the predictions of Ref. [54] for the Gaussian fixed point. The exponent for $L_c(t)$, on the other hand, appears to plateau to a value of $z \approx 3$ at high temperatures. This discrepancy is likely due to a combination of low vortex number (only $\lesssim 15$ vortices remain at such late times at high $T$) and boundary effects, although simulations in a larger system would be required to confirm this. We conclude that our results are largely consistent with the expected behaviour: the low temperature states evolve towards the non-thermal fixed point, driven by the evaporative heating of vortices, whereas the higher temperature states are subjected to dissipative vortex–phonon interactions, which drive the system towards thermalisation.

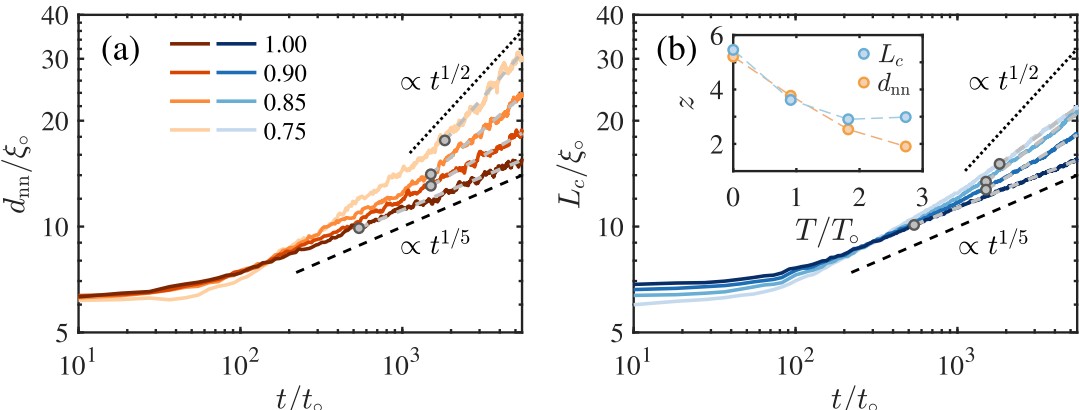

Figure 6: (a) Mean nearest-neighbour vortex distance $d_{nn}$, and (b) correlation length $L_c$, as functions of time for all four initial condensate temperatures. Power-law fits to each curve are shown as grey dashed lines, with the start point of each fit indicated by a grey dot. For comparison, the power-laws corresponding to the two fixed points, discussed in Ref. [54], are shown as dashed ($t^{1/5}$) and dotted ($t^{1/2}$) black lines in both frames. The inset of (b) shows the measured dynamical critical exponent $z$ from each fit as a function of temperature.

## 4 Conclusions and outlook

Here we have studied decaying two-dimensional quantum turbulence in Bose–Einstein condensates at finite temperatures using a classical field approach. As is the case with simpler phenomenologically damped mean-field models, the non-condensate fraction has a strong influence on the vortex dynamics. Microscopically, the non-condensate atom density modifies the condensate density in the vicinity of the vortices, and this density modulation results in a Magnus force with a component along the direction of the motion of a vortex [62, 63]. For an isolated vortex–antivortex dipole, this force component causes the pair to move closer to each other, eventually leading to their annihilation. At zero condensate temperature the Magnus force remains orthogonal to the vortex velocity vector, and therefore isolated vortex–antivortex pairs cannot annihilate. The energetics of the decaying vortex system are driven by the competition between two key mechanisms: (i) the evaporative heating of vortex dipoles that drives the system toward higher energy per vortex states and (ii) dissipative single vortex dynamics arising from the vortex 'friction', which is caused by the presence of non-condensate atoms. Ultimately, the stronger of these vortex heating and cooling effects determines the fate of the entire vortex system.

In experiments [25, 26, 51, 64], non-condensate atoms are always present. Our results imply that as long as sufficiently high condensate fraction is maintained, the qualitative results of the zero temperature Gross–Pitaevskii simulations remain valid in finite temperature systems. However, high condensate temperatures introduce dissipative effects into the vortex dynamics, resulting in cooling of the vortex gas and an erosion of vortex clustering.

In future, it will be interesting to study the cross-over from the anomalous to the thermal fixed point behaviour in further detail and to explore its potential connections to the theory of dynamical phase transitions [65]. Furthermore, the long-time evolution of the system at the coldest condensate temperatures remains an open problem—does the system eventually thermalise by reverting to the Gaussian fixed point and annihilating all vortices, as predicted? Or does the lifetime of the clusters approach infinity in the zero temperature limit? Although the vortices should gradually diffuse toward the fluid boundary due to interactions with phonons, perhaps this effect is not strong enough to overcome the topological protection of the vortices.

External forcing of a superfluid will inevitably lead to the heating of the condensate, which naturally leads to the question: is it possible to achieve driven steady state quantum turbulence, and if so, what are the properties of such a non-equilibrium system? Some of these questions for wave turbulence in three-dimensional homogeneous Bose gas have been addressed in a recent experiment by Navon *et al.* [66], in which they observed the establishment of a direct energy cascade. Similar experiments could be performed in two-dimensional systems [25, 26, 28]. In the future it will be interesting to address these questions for vortex turbulence in two-dimensions using the numerical methods utilised here. Three-dimensional simulations pose a larger numerical challenge, but could be addressed using supercomputer simulations of the GPE as utilised in Ref. [67].

## Acknowledgements

We are grateful to Thomas Billam, Thomas Gasenzer, Kristian Helmerson and Shaun Johnstone for useful discussions.

**Funding information**   We acknowledge financial support from the Australian Research Council via Discovery Programme Projects DP130102321 (T. S.), DP170104180 (T. S.), FT180100020 (T. S.). This research was also partially supported by the Australian Research Council Centre of Excellence in Future Low-Energy Electronics Technologies (project number CE170100039) and funded by the Australian Government.

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
