# Peer review of "Decaying quantum turbulence in a two-dimensional Bose-Einstein condensate at finite temperature"

_SciPost Physics, doi:SciPost Phys. 8, 039 (2020)_

## Round 1 · Referee Report · Anonymous · 2019-4-13

Strengths

1-Numerical investigation of finite-temperature effects on Onsager vortex formation in two-dimensional Bose-Einstein condensates.
2-Numerical investigation of the turbulent decay from the perspective of the rate equation of quantized vortex number.

Weaknesses

1-Validity of the stochastic projected Gross-Pitaevskii equation for the preparation of initial states.
2-Choice of the cutoff parameter in the projected GP equation.
3-Insufficient discussions on the relation between turbulent decay in the numerical simulations and nonthermal fixed points.

Report

The authors consider an interesting topic on Onsager vortex formation in two-dimensional (2D) Bose-Einstein condensates (BECs) captured by a cylindrical uniform trap. In their previous study, the authors reported that the Onsager vortex (negative temperature state) can indeed be formed in 2D BECs if the evaporative vortex cooling works well. The condition for the cooling was discussed in terms of vortex interactions and trapping-potential configurations. As a result, they found that the cylindrical uniform trap was a good platform for the Onsager vortex formation.
This paper is a sequel to the above previous works. The authors have systematically studied the Onsager vortex formation in “finite-temperature” 2D BECs by using the stochastic projected Gross-Pitaeskii (SPGP) equation for preparation of initial states and the projected GP (PGP) equation for the time evolution. In the previous work, the initial state is prepared by randomly distributed vortices, and they do not consider finite-temperature effects. The present work takes the effects into consideration by preparing the initial states with the stochastic projected GP equation. Then, investigating turbulent decay from the initial state with various temperatures, they uncover that the finite-temperature fluctuations can disturb the evaporative vortex cooling and the Onsager vortex cannot be formed in the high temperature case even if the uniform trap is used. Furthermore, time evolution of the vortex number is numerically calculated, and they discuss nonthermal fixed points (NTFPs).
Their finding for the finite-temperature effects is very useful for experimental observation of Onsager vortex, and the discussion on the NTFP is interesting since recent experiments actually succeeded in observing universal relaxation for NTFPs. However, the following two reasons prevent me from recommending the publication in the present manuscript.

Reason1) Validity of the SPGP equation and the PGP equation
The SPGP is valid if the temperature is higher than half-$T_{\rm c}$. Here, $T_{\rm c}$ is the BEC transition temperature. For example, the review paper [Advances in Physics 57 (5), 363-455 (2008)] describes this point. However, their numerical simulations consider a low temperature region $0< T/T_{\rm c}<0.3$, so that their use of the SPGP cannot be justified. Also, both SPGP and PGP methods introduce a cutoff parameter, but they have not given any discussions on how to determine it. The authors should address the issues and describe them in the revised version.

Reason2) Discussion on the relation between turbulent decay and NTFPs.
Usually, universal relaxation of NTFPs can be characterized by dynamical scaling of spatial two-pint correlation functions [Phys. Rev. Lett. 101, 041603 (2008)]. The nice review is given in [J. Berges, arXiv:1503.02907 (2015)]. Despite the importance of the scaling, the authors do not show time evolution of the correlation functions at all and there are no discussions on the dynamical scaling. In my opinion, the present results are insufficient for judging whether the turbulent decay is close to the Gaussian or anomalous nonthermal fixed points. I recommend the authors to confirm the dynamical scaling and estimate the power exponents.

If the authors satisfactory address these issues, I recommend the publication.

The followings are minor questions, which may improve the manuscript.

1-Relation between Onsager vortex formation and inverse cascades
In the introductory part, Onsager vortex formation and inverse cascades are mentioned. However, I cannot understand a relation between two concepts in the 2D GP model in the present introduction. It would be nice to clarify it.

2-Two rate equations for vortex number
Is it possible to fit the numerical data using Eq.(5)? If possible, we cannot judge which rate equations are correct. How do you answer it?

3-One-body particle loss
Realistic ultra-cold gases suffer from one-body particle loss. Does it disturb the evaporative vortex cooling?

Requested changes

1-Discuss the validity of the SPGP and PG equations.
2-Need to show time evolution of correlation functions, check the dynamical scaling, and calculate the power exponents.

  • validity: low
  • significance: good
  • originality: good
  • clarity: top
  • formatting: excellent
  • grammar: excellent

Author:  Andrew Groszek  on 2020-01-08  [id 699]

(in reply to Report 1 on 2019-04-13)

We thank the referee for their feedback and appreciate their constructive criticisms. We have updated the manuscript in the ways they have suggested, and we hope that we have addressed their concerns satisfactorily.

Requested changes:

1. “The SPGP is valid if the temperature is higher than half-Tc. Here, Tc is the BEC transition temperature. For example, the review paper [Advances in Physics 57 (5), 363-455 (2008)] describes this point. However, their numerical simulations consider a low temperature region 0<T/Tc<0.3, so that their use of the SPGP cannot be justified. Also, both SPGP and PGP methods introduce a cutoff parameter, but they have not given any discussions on how to determine it. The authors should address the issues and describe them in the revised version.” Response: While the referee states that c-field methods are only valid at finite temperature for T>~ 0.5 T_c, this is in fact a rule-of-thumb estimate based on the typical numbers of atoms in a 3D harmonic trap in alkali gas experiments. Here we consider a two-dimensional flat-bottomed trap, which has a significantly different density of states to a 3D harmonic trap, and so the temperature region of validity is different. In fact, due to the enhancement of fluctuations in lower dimensions, the validity regime is actually extended. The discussion of validity of c-field methods in the Blakie et al. review paper the referee quotes is quite detailed, and describes several conditions that are not always consistent with one another. One often-quoted validity condition for c-field methods, also discussed in the Blakie et al. review, is that the number of particles per mode should be significantly larger than one, in order that the neglection of quantum fluctuations is valid. If we normalise our system to have O(10^5) atoms in the c-field region (an experimentally feasible number), on plotting the momentum distributions for n_0 = 1 and n_0 = 0.75 [see insets of new Fig. 4(c,d)], then we can see that all the modes in all simulations in fact satisfy this condition reasonably well. For completeness, we have attached to the response an additional figure (Fig. 1 in the attachment) showing the data for n_0 = 0.9 and n_0 = 0.85 (details as for Fig 4, with n_0 = 0.9 in the left column, and n_0 = 0.85 in the right). A paragraph providing a discussion of the choice of cutoff and the validity of the c-field methodology has been added to the end of Section 2.

2. “Discussion on the relation between turbulent decay and NTFPs. Usually, universal relaxation of NTFPs can be characterized by dynamical scaling of spatial two-pint correlation functions [Phys. Rev. Lett. 101, 041603 (2008)]. The nice review is given in [J. Berges, arXiv:1503.02907 (2015)]. Despite the importance of the scaling, the authors do not show time evolution of the correlation functions at all and there are no discussions on the dynamical scaling. In my opinion, the present results are insufficient for judging whether the turbulent decay is close to the Gaussian or anomalous nonthermal fixed points. I recommend the authors to confirm the dynamical scaling and estimate the power exponents.” Response: This was an excellent suggestion. Based on this, we have performed additional analysis of the dynamical scaling of both the two-point correlation function and the mode occupation spectrum, and we have included this in the revised manuscript. We have also extracted the correlation length from the correlation function, which shows evidence of power-law scaling in time. We have added a new figure (Fig. 4) to the manuscript, which shows the collapse of correlation functions and occupation number distributions, when the axes are rescaled with the correlation length. We have also added a second subfigure to Fig. 6 (previously Fig. 5) in which the correlation length is plotted as a function of time, for comparison with the mean nearest-neighbour vortex distance. We have accordingly updated the text in Sec. 3.4 to describe the new analysis, and provide further discussions of the results. We hope our additions provide a more complete picture of the statistical evolution of the system.

Minor questions:

1. “Relation between Onsager vortex formation and inverse cascades. In the introductory part, Onsager vortex formation and inverse cascades are mentioned. However, I cannot understand a relation between two concepts in the 2D GP model in the present introduction. It would be nice to clarify it.” Response: We have added to the sentence in the introduction which draws the association between the inverse cascade and the clustering of same-sign vortices in 2D quantum turbulence.

2. “Two rate equations for vortex number. Is it possible to fit the numerical data using Eq.(5)? If possible, we cannot judge which rate equations are correct. How do you answer it?” Response: As stated in the text, these equations should be regarded as entirely phenomenological. As such, we would not claim that either model is 'correct' or 'incorrect' based on our fitting procedure, unless one provided a significantly better fit than the other. It is indeed possible to fit our data to Eq. (5) and obtain values of Gamma_1, Gamma_2, Gamma_3, Gamma_4 as a function of temperature. We state in the text that: "When fitting the two rate equations (5) and (6), we find that both describe the data equally well. However, since the latter involves fewer free parameters, we opt to use it over our earlier model." We have attached a figure to our response in order to clarify this (Fig. 2 in attachment). This figure is identical to the main frame of Fig. 3 of the manuscript, except that we have additionally overlaid the fits from Eq. (5) as red dotted lines. The values of Gamma extracted from these fits are also given in the table in the inset. As can be seen, the difference between the quality of the fits to Eq. (5) and (6) is minimal. However, the Gamma values extracted from Eq. (5) do not provide a particularly clear interpretation due to the large number of free parameters in the model. There is a general trend in the data as the temperature is changed (i.e. the four-body term dominates over the one- and two-body terms at T=0, while the opposite is true at higher temperatures); however, the picture is less clear than that provided by Eq. (6). We also emphasise that the quantitative values of Gamma should not be regarded with too much significance; rather, we are interested in the qualitative trend in the Gamma values as a function of temperature. We argue that fitting to Eq. (6) gives a clearer interpretation of how the N_v(t) curves change as a function of temperature (as seen in the insets of Fig. 3 in the text), and hence we have chosen to use that model in the manuscript.

3. “One-body particle loss Realistic ultra-cold gases suffer from one-body particle loss. Does it disturb the evaporative vortex cooling?” Response: One-body loss is expected to have a relatively unimportant effect on vortex dynamics in 2D. At the first level of approximation in a flat-trap, it will simply reduce the density, which will cause a small increase in the size of the vortex cores, and speed up the time-scale for vortex dynamics, but not have any other effect.

Attachment:

response_figures.pdf

---

## Round 1 · Referee Report · Anonymous · 2019-4-19

Strengths

1. The paper is well written and was a straightforward read for someone who works in a neighboring field, i.e., vortex dynamics of quantum fluids in three-dimensions. In my notes, I have at the top of page 5 that I was *enjoying* the read.

2. The narrative was highly logical, I walked away with a clear understanding of what the research program seeks to accomplish.

3. The references were timely and appropriate and I feel that I could *work* with/through this manuscript.

Weaknesses

1. Some of the technical terms/ideas could be unpacked a bit more. Due to the writing quality, I was able to follow the discussion without issue. However, b/c of the high quality I was left wanting more information since I expect that it would have been illuminating.

Report

Nice paper. Readable for a non-expert. It is a new result and novel to researchers in the field since it supports a mature line of research asking fundamental questions about vortex sourced dynamics in two-dimensions. For this reason, it is both interesting and important.

The quality of writing is quite high and the graphics, in the color pdf, were useful and not too difficult to decode/understand. It should be noted that in black and white (print), much of the meaning is lost and one can really only big features/trends.

The sanity checks are clear/appropriate and their results pass them.

That said, I do have some suggestions that could improve the document, which I discuss below.

Requested changes

1. The inline nonzero moment defined just after Eq. (3) is confusing since the romanized d seems out of place. If the \hbars from Eq. (3) were written with the dt terms, then the reader would quickly understand the notation associated with the inline equation. You could also write the dt/\hbar after the delta.

2. The reader could be helped with the inclusion of a sentence in the introduction just following the two sentences about PGPE and SPGPE. At this point in my read, I now know your tools and your focus. However, if I am not familiar with the tools, then I won't really see the contrast between them and the classical field approach. A summarizing sentence would set the reader up for the discussion of your methodology.

3. I understand why 2.2 and 2.3 are separated out from 2.1 but as single paragraphs, they dangle. Perhaps it would be better to just incorporate them?

4. In 2.2, is there a reason for working with the region defined by 95% of the radius? I didn't find intuition with a quick skim backward. I have several ideas about why one might/should do this but it would be good to know the author's perspective.

5. The start of 3.2 and its footnote was dense/technical. Consider reformatting into a trichotomized list. The technical matters could be unpacked more but in the end, the reader needs to walk away with the three regimes. This gets a little lost in the succinct but technical discussion which left me feeling split between re-reading the technical statements or pushing forward with the trichotomy in mind.

6. I found the first sentence of 3.4 awkward and felt that it could be clearer with a re-organization of content to get rid of several parenthetical statements.

7. In the second to last paragraph of section 3, the authors state that the system may eventually return to the Gaussian fixed point. Why does this argument exist? It seems to me that clusters might have a similar sort of topological protection and so I could imagine that they may not totally annihilate each other, at least at zero temperature as the authors note. This idea seems to split over this paragraph and two others in section 4. As it is an important and interesting question, perhaps a more focused discussion would be better. Alternatively, the authors could add on a sentence that hints at the forces at play that would prevent long-time protection of the clusters and the openness of this question.

8. I like the rapid-fire question list at the end of the conclusion. It really helps punctuate their program/results. However, since I was enjoying the read, I wanted to hear about the author's opinions about whether the methods of the paper are enough to address these questions. If not, then what changes/hurdle do they expect? Completing in this way would keep me interested in their next work or becoming more acquainted with this/their research program.

  • validity: good
  • significance: good
  • originality: high
  • clarity: high
  • formatting: good
  • grammar: excellent

Author:  Andrew Groszek  on 2020-01-08  [id 698]

(in reply to Report 2 on 2019-04-19)
Category:
answer to question

We thank the referee for their overall positive comments and feedback. Responses to all of their suggestions and queries can be found below.

Requested changes:

1. “The inline nonzero moment defined just after Eq. (3) is confusing since the romanized d seems out of place. If the hbars from Eq. (3) were written with the dt terms, then the reader would quickly understand the notation associated with the inline equation. You could also write the dt/hbar after the delta.” Response: As suggested, we have moved the dt term to the end of the in-line expression. We agree that this improves the clarity.

2. “The reader could be helped with the inclusion of a sentence in the introduction just following the two sentences about PGPE and SPGPE. At this point in my read, I now know your tools and your focus. However, if I am not familiar with the tools, then I won't really see the contrast between them and the classical field approach. A summarizing sentence would set the reader up for the discussion of your methodology.” Response: We have added a brief description of the c-field methodology, and the PGPE and SPGPE in the introduction as requested.

3. “I understand why 2.2 and 2.3 are separated out from 2.1 but as single paragraphs, they dangle. Perhaps it would be better to just incorporate them?” Response: Thank you for the suggestion, and we have considered this. However, following the report of the second referee, we have extended 2.3, and so have decided to retain the current structure.

4. “In 2.2, is there a reason for working with the region defined by 95% of the radius? I didn't find intuition with a quick skim backward. I have several ideas about why one might/should do this but it would be good to know the author's perspective.” Response: We only consider vortices in the region defined by 95% of the radius due to numerical considerations. In the low-density regions near the boundary of the system, the phase can become filled with numerical singularities (‘ghost’ vortices) that are picked up by our vortex detection algorithm. These ghost vortices do not enter the system or affect the dynamics in a meaningful way, so we therefore remove them from consideration by ignoring any singularities detected in the region r > 0.95*R. We have included this explanation in the text.

5. “The start of 3.2 and its footnote was dense/technical. Consider reformatting into a trichotomized list. The technical matters could be unpacked more but in the end, the reader needs to walk away with the three regimes. This gets a little lost in the succinct but technical discussion which left me feeling split between re-reading the technical statements or pushing forward with the trichotomy in mind.” Response: We appreciate the suggestion to reformat the start of Sec. 3.2 as a trichotomised list, and to expand on the explanation therein. We have updated the text, and hope that the readability of this section has been improved as a result.

6. “I found the first sentence of 3.4 awkward and felt that it could be clearer with a re-organization of content to get rid of several parenthetical statements.” Response: We have rewritten the opening sentences of Sec. 3.4, and hope that they are now clearer than they were previously.

7. “In the second to last paragraph of section 3, the authors state that the system may eventually return to the Gaussian fixed point. Why does this argument exist? It seems to me that clusters might have a similar sort of topological protection and so I could imagine that they may not totally annihilate each other, at least at zero temperature as the authors note. This idea seems to split over this paragraph and two others in section 4. As it is an important and interesting question, perhaps a more focused discussion would be better. Alternatively, the authors could add on a sentence that hints at the forces at play that would prevent long-time protection of the clusters and the openness of this question.” Response: To clarify the prediction that the system should eventually return to the Gaussian fixed point, we have added the following text to the end of Sec. 3.4.2: "It is predicted that the system would eventually return to the Gaussian fixed point, because vortex–phonon interactions give rise to gradual vortex diffusion [61], which serves to break up vortex clusters and encourage vortex–antivortex annihilation. However, it is possible that the time required for the system to cross over to the Gaussian fixed point may approach infinity as T->0." Naively, one would expect that a nonlinear equation such as the Gross-Pitaevskii model should exhibit ergodicity, and therefore eventually 'forget' the details of its initial state and thermalise by finding the maximum entropy state available under the constraint of energy and norm conservation. In this case, the thermal state would correspond to a system without vortices, as this would maximise the entropy. However, it is possible that the topological protection of the vortex clusters would prevent the system from ever reaching this state at the lowest temperatures. We have also added the following sentence in the conclusion to make this point: "Although the vortices should gradually diffuse toward the fluid boundary due to interactions with phonons, perhaps this effect is not strong enough to overcome the topological protection of the vortices."

8. “I like the rapid-fire question list at the end of the conclusion. It really helps punctuate their program/results. However, since I was enjoying the read, I wanted to hear about the author's opinions about whether the methods of the paper are enough to address these questions. If not, then what changes/hurdle do they expect? Completing in this way would keep me interested in their next work or becoming more acquainted with this/their research program.” Response: Thank you for the suggestions. We have expanded the conclusion section to elaborate on some of our questions and comments, and to incorporate some recent work. We hope the referee finds these additions valuable.

---

## Round 2 · Referee Report · Anonymous (Referee 1) · 2020-1-30

Report

The authors have satisfactorily revised the manuscript to answer all my questions and comments, and thus I am happy to recommend the publication in the current form.

---

## Round 2 · Author Response

We greatly appreciate the suggestions made by the referees, and we believe the presentation of our results has improved considerably based on the feedback provided in their reports. We hope that the revised manuscript is deemed suitable for publication.

---

## Round 2 · List of Changes

Warnings issued while processing user-supplied markup:

  • Inconsistency: Markdown and reStructuredText syntaxes are mixed. Markdown will be used.
    Add "#coerce:reST" or "#coerce:plain" as the first line of your text to force reStructuredText or no markup.
    You may also contact the helpdesk if the formatting is incorrect and you are unable to edit your text.

========================== Changes based on Report 1: ========================== - Sec. 1, par. 2, sentence 2: added "...as these two phenomena are both characterised by the emergence of system-scale eddies."

  • Sec. 2.3: paragraph 2 added to describe validity of c-field methods.

  • Sec. 3.4 has been restructured in order to include the additional analysis suggested by the referee, and the corresponding data is shown in the new Fig. 4 and the new Fig. 6(b) (including its inset; the caption of Fig. 6 has been updated accordingly). An entirely new Sec. 3.4.1 has been added to describe the additional data, while the previous discussion has been split over what is now Secs. 3.4.2 and 3.4.3. While the text in Sec. 3.4.2 is mostly unchanged from the previous version, Sec. 3.4.3 has been largely rewritten to accommodate the newly added results. A second paragraph has also been added to the beginning of Sec. 3.4 to summarise the subsequent subsections.

========================== Changes based on Report 2: ========================== - Sec. 1, par. 4. Replaced sentences: "The unitary projected Gross–Pitaevskii equation (PGPE) that conserves both the energy and the normalisation of the classical field is used for describing the dynamics of the Bose gas. We systematically vary the initial condensate temperature by sampling microstates using the stochastic projected Gross–Pitaevskii equation (SPGPE) and determine the resulting effect on the turbulent dynamics for an ensemble of statistically equivalent vortex distributions." with: "Briefly, this uses the Gross-Pitaevskii equation to simulate the dynamics of not only the condensate, but also the low-energy thermal fluctuations of the field. We simulate the grand canonical stochastic projected Gross–Pitaevskii equation (SPGPE), describing the classical field coupled to a bath, to generate initial thermal ensembles. We imprint vortices on these microstates to form an ensemble of vortex distributions, and determine the resulting effect of the finite temperature on the turbulent vortex dynamics by integrating the microcanoncial projected Gross–Pitaevskii equation (PGPE) that conserves both energy and normalisation of the classical field."

  • In the text following Eq. (3), the factor of dt has been moved to the end of the expression for the non-zero moment of the driving noise.

  • Sec. 2.2, final sentence: added "in order to avoid the detection of numerical 'ghost' vortices in regions of low classical field density" (and associated reference)

  • Sec. 3.2, par. 1: the description of the three vortex temperature regimes has been formatted as a list, and each item has been expanded to include additional details. The footnote has been incorporated into the main text.

  • Sec. 3.4, par. 1 has been reworded as per the referee's request. It now reads: "Despite the complexity of 2D QT, it has been demonstrated that the dynamics can in many cases be characterised in terms of statistically steady distributions that are only weakly dependent on the microscopic details of the system. This is a general and powerful approach to understanding the evolution of a system out of equilibrium, and allows its characterisation in terms of far from equilibrium universality classes [56,57]."

  • Sec 3.4.2, final paragraph: the final sentence, which previously read: "It is predicted that the system would eventually return to the Gaussian fixed point; although the time required to do so may approach infinity as T->0" has now been changed to: "It is predicted that the system would eventually return to the Gaussian fixed point, because vortex–phonon interactions give rise to gradual vortex diusion [61], which serves to break up vortex clusters and encourage vortex–antivortex annihilation. However, it is possible that the time required for the system to cross over to the Gaussian fixed point may approach infinity as T->0."

  • Sec. 4, final paragraph: the question "does the system eventually revert to the Gaussian fixed point by annihilating all vortices?" has been changed to: "does the system eventually thermalise by reverting to the Gaussian fixed point and annihilating all vortices, as predicted?" Immediately after this, the following sentence has been added: "Although the vortices should gradually diffuse toward the fluid boundary due to interactions with phonons, perhaps this effect is not strong enough to overcome the topological protection of the vortices."

  • Sec. 4, final paragraph: Four new sentences have been added to the end of the section: "Some of these questions for wave turbulence in three-dimensional homogeneous Bose gas have been addressed in a recent experiment by Navon et al. [66], in which they observed the establishment of a direct energy cascade. Similar experiments could be performed in two-dimensional systems [25,26,28]. In the future it will be interesting to address these questions for vortex turbulence in two-dimensions using the numerical methods utilised here. Three-dimensional simulations pose a larger numerical challenge, but could be addressed using supercomputer simulations of the GPE as utilised in Ref. [67]"

========================== Other changes: ========================== - Added references [23, 41, 58, 61, 66, 67].

  • Removed the final sentence from the abstract. It previously read: "The cross-over between these two dynamical behaviours is found to occur at earlier times with increasing condensate temperature."

Sec. 1:

  • Par. 3, sentence 2: added the words "incompressible kinetic" before the word "energy".

  • Par. 4 has been split into two paragraphs. Added the words "In our simulations" to the beginning of the new paragraph 5.

  • Final paragraph: based on changes made as a result of Report 1, we have replaced the sentence: "The time-dependence of the nearest-neighbour vortex spacing provides evidence of a cross-over from evolution towards an anomalous non-thermal fixed point to a thermal fixed point." to: "Evidence is also provided for universal scaling in our simulations, and based on this we are able to interpret the dynamics as evolving towards either a thermal or non-thermal fixed point, depending on the temperature of the system."

Sec. 2.1:

  • Par. 2, sentence 3: removed factor of 10^-5 in front of definition of the unit T_o. The numerical temperature had previously been inadvertantly divided by a factor of N=10^5 (the normalisation of the classical field). This has now been corrected, and as a result the factor of 10^-5 has been removed. This change does not affect any other temperature values quoted in the manuscript.

  • Par. 2: new sentence added at the end of the paragraph to express the numerical units in terms of experimentally realistic quantities.

  • Par. 3, sentence 2: the words "above the cutoff" have been added.

  • Par. 4, sentence 1: the words "damping parameter" have been replaced with "growth coefficient", and the word "although" has been removed from the beginning of the comment in parantheses.

  • Par. 4, sentence 2: we have changed: "the temperature is set to a constant value in the range 0 < T/T_o <~ 3 throughout each simulation" to: "the temperature is set to one of three values, T/T_o =~ {0.9, 1.8, 2.7}, for each simulation ensemble." Immediately following this, we have added the sentence: "Note that we also perform a T = 0 simulation, but we do not need to evolve the SPGPE (3) to find the initial state."

  • Par. 4, sentence 4: the words "the condensate fraction vanishes in equilibrium" have been replaced with "the condensate fraction of the classical field vanishes in equilibrium while holding the chemical potential constant". Following this sentence, we have added: "Note that this is different to the typical scenario in which the total number of atoms is kept constant as the temperature is varied."

  • Par. 6, first sentence: the word "extracted" has been replaced with "determined".

  • Final paragraph: immediately after defining the function \chi(r), the word "captures" has been replaced with "approximates".

Sec. 2.2

  • First sentence: added "(corresponding to t~3.8s for the physical parameters chosen in Sec. 2.1)"

Sec. 2.3

  • The two paragraphs that previously comprised Sec. 2.3 have been joined into a single paragraph, and following "\Delta x ~ \xi_o / 3.5", we have added: "which ensures that the vortex cores are accurately represented by the numerics. This choice leads to a value for the wavevector k_cut of the projector."

Sec. 3.1

  • First sentence: corrected the reference to "Fig. 1(d)-(e)" to "Fig. 1(d)-(f)".

Sec. 3.2

  • Par. 1, first sentence: the words "incompressible kinetic" have been added before "energy".

  • Par. 1, sentence 2 has been added: "The inverse temperature is defined as \beta = 1 / k_b \partial S / \partial E", where E and S are the entropy and energy of the vortex gas, respectively."

  • Par. 1, immediately following the list: the words "extract this" have been replaced with "determine the".

  • Par. 2, sentence 1: the word "extracted" has been replaced with "determined".

  • Par. 2, sentence 2: the words "vortex gas evaporatively heats up" have been replaced with "evaporative heating of the vortex gas proceeds"

Sec. 3.3

  • Par. 1, first sentence: the words "occasionally also" have been reversed to "also occasionally".

  • Fig. 3 caption, first sentence: "decay curves" has been replaced with "as a function of time".

Sec. 3.4.2

  • Par. 1, sentence 1: the words "Previously, Groszek et al. identified related scale invariant behaviour" have been changed to: "Previously, Groszek et al. identified scale invariant behaviour in decaying quantum turbulence".

  • Par. 1, sentence 2: the words "the decaying quantum turbulence rapidly approaches" have been replaced with "the vortex configuration rapidly approaches".

---

## Editorial Decision

published